# Harmful Alcohol and Drug Use Is Associated with Syndemic Risk Factors among Female Sex Workers in Nairobi, Kenya

**DOI:** 10.3390/ijerph19127294

**Published:** 2022-06-14

**Authors:** Alicja Beksinska, Emily Nyariki, Rhoda Kabuti, Mary Kungu, Hellen Babu, Pooja Shah, Chrispo Nyabuto, Monica Okumu, Anne Mahero, Pauline Ngurukiri, Zaina Jama, Erastus Irungu, Wendy Adhiambo, Peter Muthoga, Rupert Kaul, Janet Seeley, Helen A. Weiss, Joshua Kimani, Tara S. Beattie

**Affiliations:** 1Department of Global Health and Development, London School of Hygiene and Tropical Medicine, London WC1H 9SH, UK; pooja.shah@lshtm.ac.uk (P.S.); janet.seeley@lshtm.ac.uk (J.S.); tara.beattie@lshtm.ac.uk (T.S.B.); 2Partners for Health and Development in Africa (PHDA), UNITID, College of Health Sciences, Nairobi P.O. Box 30197-00100, Kenya; nyarikiemily@gmail.com (E.N.); rhodakabuti@gmail.com (R.K.); kunguwmary@gmail.com (M.K.); hellenbabu@gmail.com (H.B.); chrisponyamweya36@gmail.com (C.N.); okumumonica9@gmail.com (M.O.); amahero@csrtkenya.org (A.M.); polycarpngugi@gmail.com (P.N.); zjama@phdaf.org (Z.J.); eirungu@phdaf.org (E.I.); wendyadhiambo@yahoo.com (W.A.); pmuthoga@phdaf.org (P.M.); jkimani@csrtkenya.org (J.K.); 3Department of Medicine, University of Toronto, Toronto, ON M5S 1A1, Canada; rupert.kaul@utoronto.ca; 4MRC International Statistics & Epidemiology Group, Department of Infectious Disease Epidemiology, London School of Hygiene and Tropical Medicine, London WC1E 7HT, UK; helen.weiss@lshtm.ac.uk

**Keywords:** female sex workers, alcohol, harmful alcohol use, drugs, harmful substance use, cannabis, amphetamines, Kenya, quantitative methods, qualitative methods

## Abstract

Background: Female Sex Workers (FSWs) are at high risk of harmful alcohol and other drug use. We use quantitative data to describe the prevalence of alcohol and other drug use and identify associated occupational and socio-economic risk factors, and aim to elucidate patterns of alcohol and drug use through information drawn from qualitative data. Methods: Maisha Fiti was a mixed-method longitudinal study conducted in 2019 among a random sample of FSWs in Nairobi, Kenya. We used baseline date from the behavioural–biological survey, which included the WHO Alcohol, Smoking and Substance Involvement Screening Test that measures harmful alcohol and other drug use in the past three months (moderate/high risk score: >11 for alcohol; >4 for other drugs). In-depth interviews were conducted with 40 randomly selected FSWs. Findings: Of 1003 participants, 29.9% (95%CI 27.0–32.6%) reported harmful (moderate/high risk) alcohol use, 21.5% harmful amphetamine use (95%CI 19.1–24.1%) and 16.9% harmful cannabis use (95%CI 14.7–19.2%). Quantitative analysis found that harmful alcohol, cannabis and amphetamine use were associated with differing risk factors including higher Adverse Childhood Experience (ACE) scores, street homelessness, food insecurity (recent hunger), recent violence from clients, reduced condom use, depression/anxiety and police arrest. Qualitative interviews found that childhood neglect and violence were drivers of entry into sex work and alcohol use, and that alcohol and cannabis helped women cope with sex work. Conclusions: There is a need for individual and structural-level interventions, tailored for FSWs, to address harmful alcohol and other drug use and associated syndemic risks including ACEs, violence and sexual risk behaviours.

## 1. Introduction

Harmful alcohol and other drug use are major public health concerns, contributing to 1.5% of the global burden of disease [1] and increasing the risk of many non-communicable and infectious diseases [2]. In recent years, alcohol use has been increasing in many low- and low-middle-income countries (LMICs) [3,4], with consumption and risks higher in certain key populations such as sex workers [5]. Sex work, defined as the receipt of money or goods in exchange for sexual services, is criminalised in most parts of the world [6,7]. Female Sex Workers (FSWs) face occupational risks including violence from clients, and high levels of HIV and other sexually transmitted infections (STIs) as well as structural inequalities including police arrest, discrimination, poverty, and gender inequality [6,7]. FSWs often use alcohol and other drugs due to their wide availability in the sex work industry [8], as part of socialising with clients, and to cope with the daily challenges of sex work [9]. These factors may predispose FSWs to increased risks of harmful alcohol and other drug use.

Social, cultural and economic factors have an effect on the structural and occupational risks associated with sex work [7]. Due to global differences in levels of sex work criminalisation and socio-cultural attitudes to alcohol use, alcohol and other drug use and associated risks are likely to differ for sex workers in LMICs compared to those in high-income countries. A recent meta-analysis of alcohol and other drug use among FSWs in LMICs estimated a pooled prevalence for harmful alcohol use of 40%, with most studies using the Alcohol Use Disorders Identification Test (AUDIT) (cut off score > 8), any other drug use in the last 6 months of 24.4% [10], and injecting drug use 15.3%. Estimates of specific drug use among FSWs range widely across different settings with prevalence ranging from 16.5–80% for recent cannabis use and 5.4–98.6% for recent methamphetamine use [10] (Karlsen et al., personal communication). No previous studies reporting drug use, other than alcohol, among FSWs in LMICs have used validated measurement tools.

Among FSWs alcohol use is associated with increased sexual risk behaviours including reduced condom use [8,11,12,13,14,15], increased prevalence of HIV and STIs [16,17] as well as reduced uptake and adherence to HIV treatment [8,18]. Alcohol use has been associated with increased risk of experiencing violence [8,16], police arrest, and income loss during sex work [19]. A recent systematic review of mental health among FSWs in LMICs found significant associations between alcohol use and recent suicidal behaviour, and illicit drug use and depression [20]. The impacts of adverse early life experiences [21] such as violence and neglect in childhood on adult alcohol and other drug use are well understood in the general population. Overlapping health and social risk factors including childhood adversity, violence, risky sexual behaviours and harmful alcohol or other drug use may be considered from a ‘syndemics’ perspective, as a clustering of risks [22] which perpetuate each other across the life course. There is a need to better understand these risk pathways, including early-life risk factors, in order to prevent and treat harmful alcohol and other drug use among FSWs.

In Kenya, approximately one in four (21.3–26.8%) FSWs report harmful or hazardous alcohol use (AUDIT) [16,23,24]. Reports of drug use among FSWs include ‘drug use’ in the past 6 months (34.1%) [25], ‘substance use’ in the last 6 months (34.5%) with high frequency of ‘khat’ use (30.2%) [26], and daily marijuana use in the last month (12.7%) [27]. Heroin use is increasing in Kenya [28] and previous research in Kenya has indicated an emerging overlap between injecting drug use and sex work [18,29]. A major limitation in assessing the burden of harmful drug use among FSWs in Kenya and globally is that no studies to date have used validated tools, with most studies using variable time frames, not reporting on specific drugs, or failing to distinguish between any vs. harmful use. Although alcohol use has been more accurately measured, risk factors are less well studied in this population. Given that alcohol and other drugs are often used together [16,30] this makes it challenging to design interventions, as the scale or severity of drug use is not adequately understood.

Maisha Fiti is a mixed-method, longitudinal study with FSWs in Nairobi, which aims to examine associations between violence, mental health, harmful alcohol and other drug use, biological changes to the immune system, and HIV/STI prevalence and risk factors. In this paper, we use quantitative data collection methods, using validated tools, to examine harmful alcohol and other drug use and associated risk factors across the life course. Through qualitative data collected in the same study, we illustrate the place of alcohol and drug use in women’s entry into and subsequently during sex work.

## 2. Materials and Methods

### 2.1. Study Design and Setting

We used baseline cross-sectional quantitative and qualitative data collected from June-December 2019 as part of the Maisha Fiti study. The study was designed in consultation with FSWs in Nairobi, and with community mobilisers and staff working at seven Sex Worker Outreach Programme (SWOP) clinics located across Nairobi County. These clinics deliver their services with the support of Partners for Health and Development Africa (PHDA) who provide HIV/STI prevention and treatment services to FSWs across Nairobi. Further details of how SWOP clinics recruit FSWs through peer education and micro-planning has previously been reported in detail [31].

Stakeholder meetings were held with Sex Worker Community Based Organisations at the study design stage, and their input helped inform the study design and participant recruitment. During the study team training, half of the team were peer sex worker educators and their input helped to inform the study tools and their administration. At the completion of the study team training, the team piloted the survey tools with volunteer respondents from the sex worker community, and the questionnaires were amended following their feedback.

Sample size calculations and sampling methodology have been described in detail previously [32]. In summary, sample calculations were as follows: SWOP clinic data has reported approximately 45% of FSWs have problems with alcohol (determined using CAGE scores). Assuming 1:1 exposure of problem alcohol use (50% exposed vs. 50% unexposed), a sample of 670 HIV negative women provides power to detect a 10% difference in the proportion of women who have genital inflammation (25% vs. 15%) at 90% power [33]. HIV prevalence among FSWs attending SWOP clinics in Nairobi is approximately 25%, therefore the target sample size was 1000 FSWs.

Eligibility criteria for Maisha Fiti were: Women aged 18–45 years, who were in active sex work, who were not pregnant or breast-feeding, and with no underlying chronic illness that could affect their immunology other than HIV. When we refer to FSWs, this includes women who were assigned female at birth. Transgender women were not included in the study as their risks and experiences of sex work are considered to be significantly different.

An estimated 39,000 women sell sex in Nairobi County, Kenya [34]. All women attending SWOP clinics have a unique clinic barcode. Potential participants were selected from the clinic attendance lists of all FSWs who had attended a SWOP clinic in the previous 12 months and who met the eligibility criteria, using random sampling proportional to clinic size. In total, from a sampling frame of 10,292 FSWs, 1200 FSWs were randomly sampled, of whom 1039 were eligible, and 1003 agreed to participate.

Women interested in participating were given an appointment at the dedicated study clinic where they were screened for eligibility and received detailed information about the study. For women with low literacy, information was read to them by study staff. Consenting eligible participants completed a behavioural–biological survey. In addition, 40 participants were randomly selected during enrolment to participate in a qualitative in-depth interview (IDI).

Participants with mental health or alcohol/other drug use problems were referred to a trained psychological counsellor at the study clinic for assessment and treatment. All women who tested positive for HIV during the study were counselled and encouraged to enrol in HIV care. All women who tested positive for STIs were offered appropriate treatment free of charge.

The study design, process and data collection have previously been described in detail [32,35].

### 2.2. Data Collection and Study Procedures

#### 2.2.1. Quantitative Data

The validated WHO ASSIST tool was used to measure alcohol and other drug use in the previous three months, including amphetamines, cannabis, cocaine, hallucinogens, sedatives and inhalants [36]. The ASSIST items have shown good test–retest reliability for each substance, with average kappa coefficients ranging from 0.61 to 0.78 [37]. Women were considered to have harmful alcohol (cut-off scores: moderate risk > 11; high risk > 27) or other drug (cut-off scores: moderate risk > 4; high risk > 27) use if they scored as either moderate or high risk, which guides the need for clinical interventions [36]. High risk use indicates drug or alcohol dependence.

ACEs were measured using the WHO Adverse Childhood Experiences International Questionnaire (ACE-IQ) [38] (Cronbach’s alpha = 0.85) [39]. Due to the length of the questionnaire, three questions from the WHO ACE-IQ tool were not included (bullying from peers, and emotional and physical neglect from parents/guardians). An additional question about street homelessness in childhood was incorporated as it was considered relevant for this population. We examined individual associations of specific ACEs with the outcomes and created a three-item categorical ACE score variable (≤4; 5–8; 9–12) which comprised 12 components (1—household member depressed/institutionalised/suicidal; 2—household member misused alcohol/substances; 3—household member in prison; 4—parents divorced/separated; 5—parent/guardian died; 6—witnessed emotional/physical violence between household members; 7—lived on the street in childhood; 8—experienced emotional violence; 9—experienced sexual violence; 10—experienced physical violence; 11—witnessed violence in the community; 12—experienced displacement/destruction of home/violence during war). This approach was guided by a systematic review and meta-analysis that reports ACEs have a cumulative impact on subsequent mental health problems [21] and on the WHO ACE-IQ scoring tool [38].

Mental health problems were measured using the Patient Health Questionnaire (PHQ-9) for depression (score ≥ 15 = moderate/severe depressive disorder) (Cronbach’s α = 0.89, sensitivity  =  88%, specificity = 88%) [40]; the Generalised Anxiety Disorder (GAD-7) tool for anxiety (score ≥ 10 = moderate or severe anxiety) (Cronbach’s α = 0.92, sensitivity = 89%, specificity  =  82%) [41] and the Harvard Trauma Questionnaire (HTQ-17) for PTSD (score ≥ 2.5 positive for PTSD) (Cronbach’s α  =  0.8–0.9) [42]. Recent suicidal behaviour included recent suicidal ideation and/or recent suicide attempt, which were defined as ‘having thoughts about ending your life in the last 30 days’ and having ‘attempted to end your life in the last 30 days’, respectively.

The WHO Violence Against Women 13-item questionnaire that measures frequency and severity of Intimate Partner Violence (IPV) was adapted to include violence perpetrated by non-IPs (e.g., police, strangers, clients, etc.) [43]. We asked about violence experiences in the past six months and at any time.

#### 2.2.2. Laboratory Methods

Urine samples were collected for pregnancy testing, and to test for Neisseria gonorrhoeae (NG) and Chlamydia trachomatis (CT) (using Gene Expert Assay). Blood was collected for Treponema pallidum (syphilis) diagnostics using the rapid plasma reagin assay. HIV status was screened with rapid tests, with confirmation using HIV DNA Genexpert. Self-collected vaginal swabs were used to diagnose Trichomonas vaginalis (TV; OSOM Trichomonas Rapid Test; SEKISUI Diagnostics, LLC) and Bacterial Vaginosis (BV; Gram’s stain and Nugent scoring).

#### 2.2.3. Qualitative Data

IDIs were conducted using semi-structured guides in English and Kiswahili. The guiding questions included: background characteristics and entry into sex work, experiences of alcohol and other drug use, experience of violence, mental health and financial situation. Interviews were conducted by two qualitative researchers in protected spaces at the study clinic. All interviews were face to face and lasted between 60–80 min. Interview notes including observable nuances were written up. All interviews except for one were audio recorded, transcribed and translated.

### 2.3. Integration of Mixed Methods and Conceptual Framework

The overall aims of this study were to describe the prevalence of alcohol and other drug use, and to identify risk factors associated with alcohol and other drug use. In addition, we aimed to elucidate patterns of alcohol and drug use through information drawn from our qualitative data. We developed a conceptual framework (Figure 1) using an eco-social life course theory approach [44]. The approach was informed by the structural determinants framework for sex work developed by Shannon et al. to explore the relationships between sociodemographics, sex work characteristics, HIV/STI risk, violence experience, mental health, and harmful alcohol and other drug use [45,46]. This conceptual framework was used to inform analysis and guide the integration of mixed-method results. During analysis, we used qualitative narratives to inform our understanding of the quantitative findings and we report on these using an integrated approach in the results.

### 2.4. Analyses

#### 2.4.1. Quantitative Analyses

Data were double-entered and statistical analyses were conducted in STATA 16.1 (Stata Inc., College Station, TX, USA). Although women < 25 years old represented 11.7% of women eligible for the study, of the 1003 women enrolled in the study, 21.1% were <25 years. Therefore sampling weights were generated and the dataset was appropriately weighted to account for age during analysis. The conceptual framework in Figure 1 guided analysis. Associations were estimated using odds ratios (OR), with *p*-values obtained using a joint hypothesis test via the adjusted Wald test (to allow for sampling weights). We used a hierarchical modelling approach [47] to build multivariable models for each outcome based on our conceptual framework, which describes the hierarchical relationship between distal and proximal determinants of harmful alcohol and drug use. Level 1 variables included a core set of distal sociodemographic variables and early life risk factors, including age, socio-economic status (SES), religion, literacy (a marker of education level, which was assessed by asking women during the behavioural questionnaire to read a sentence), as well as ACEs. Hierarchical modelling involves incorporating temporal, biological and social understandings about the relationships between determinants of disease, taking into account the effect of mediating variables.

The overall effect of level 1 variables (Figure 1) were examined in Model 1 adjusted for other level 1 variables, but not level 2 or 3 variables as these represent mediating factors. In Model 2, level 2 variables were examined adjusted for all level 1 and 2 variables. Model 3a and 3b adjusted for levels 1, 2 and either 3a or 3b variables. All variables in each level (shown in the conceptual framework) were included in the corresponding and subsequent model, i.e., all level 1 variables were adjusted for in model 2, 3a and b regardless of whether they were significant in model 1, however, we only report associations with *p* < 0.1 in results. To adjust for clustering by clinic, SWOP clinic was included as a fixed effect in all models.

#### 2.4.2. Qualitative Analyses

Data were thematically analysed, a process which began after the completion of the first interviews. Debriefing meetings were held after every two to four interviews to enable the research team (TB, PS, RK, JK, MK, HB & EN) to discuss the findings and suggest areas for follow-up. These discussions also included clarifications from preceding interviews, as well as identifying emerging themes and areas for follow-up and clarification interviews. A coding framework was developed by four members of the research team (TB, PS, RK & EN) who were tasked with reading and re-reading the interview notes and scripts in their entirety. The adequacy of interview notes for analysis has been considered in qualitative studies [48]. Continual engagement with the data allowed the analysis team to gain an overview of emerging themes, and the confidence that there were adequate data to develop a robust and valid understanding of these. Within the analysis used in this paper the key themes were: 1. Women’s contextual factors and alcohol and substance use; 2. Reasons for consuming alcohol and using substances in sex work; 3. Impacts of alcohol and substance use on sex work; 4. Impacts of harmful alcohol use and drugs on family relations, individual physical and mental health. The interviews were uploaded to NVivo 12 software (QSR International Melbourne, Australia) for coding analysis. For a uniform understanding of the coding framework, one script was randomly picked and coded by the four members for agreement. The audio-recorded interviews were transcribed verbatim and translated into English language where Kiswahili language had been used. Five transcripts were independently coded by the team and no further topics were identified. The remaining transcripts were distributed amongst the team members and coded individually. For this paper, analytical memos were developed to synthesise the women’s experiences of alcohol and other drug use.

#### 2.4.3. Ethical Considerations

The Maisha Fiti study was approved by the Kenyatta National Hospital Ethics and Research Committee, the London School of Hygiene and Tropical Medicine (LSHTM) Ethics Committee and the University of Toronto ethics committee.

## 3. Results

### 3.1. Sample Demographics

Of the 1039 eligible FSWs, 1003 (96.5%) consented and participated in the baseline survey. The median age was 32 years (range 18–45 years). Most women had previously been married (81.2%) (Table 1), but over two thirds (69.2%) were currently living with children and no partner, with just 6.8% living with a male partner. Less than half (43.7%) reported additional income other than sex work and 33.9% reported missing a meal in the last week due to financial constraints. The mean age at first sex was 16.3 years with 5.4% of women reporting sexual debut < 13 years. Most women (91.4%) worked from a lodge/hotel/rented room, with a median client volume per week of 3 (range 0–70). HIV prevalence was 28.0% (95%CI: 25.3–30.9%) and chlamydia was the most prevalent STI (5.7%; 95%CI: 4.5–7.2%).

Prevalence of any reported harmful alcohol or other drug use (moderate/high risk ASSIST score) was 44.2% (95%CI: 41.2–47.2%). Harmful alcohol use was most prevalent (29.8%; 95%CI: 27.0–32.6%) followed by amphetamine (21.5%; 95%CI: 19.1–24.1%) and cannabis (16.8%; 95%CI: 14.7–19.2%) use, with 6.5% of women reporting harmful use of all three drugs (Figure 2). Overall, 11.2% of participants reported high-risk (dependent) use, with high-risk alcohol use being most common (9.4%). Prevalence of other harmful drug use was low (sedatives 2.1%, cocaine 0.3%, opioids 0.5%, inhalants 0.1% and hallucinogens 0.1%). Only four (0.5%) women reported injecting drug use in the last three months.

### 3.2. Adverse Childhood Experiences

Women with harmful alcohol use were significantly more likely to have a higher ACE score (ACE score 9–12: aOR = 3.66; 95%CI: 2.34–5.72; *p* < 0.001), to have lived on the street (aOR = 1.95; 95%CI: 1.31–2.92; *p* = 0.001), and to have experienced sexual/physical violence (aOR: 3.27; 95%CI: 2.13–5.02; *p* < 0.001) in childhood, compared with women without harmful alcohol use (Table 2, model 1). Similarly, women with harmful cannabis use were more likely to report higher levels of ACEs (ACE score 9–12: aOR= 2.99; 95%CI: 1.72–5.21; *p* < 0.001), to have lived on the street, and to have experienced sexual/physical violence in childhood (Table 2, model 1). Women with harmful amphetamine use were more likely to report having lived on the street, but amphetamine use was not associated with higher ACE levels overall.

These findings are reflected in the qualitative interviews in women’s accounts of their childhood experiences of poverty, neglect, teenage pregnancies, early marriages and relationship breakdowns, leading to entry into sex work and use of alcohol and other drugs for survival. A 29-year-old mother of two, who was abandoned by her own mother at an early age, and raised and educated by her grandmother, became pregnant at age 16 and married the father of her unborn baby due to household poverty. This marriage did not last as the man was abusive, forcing her to escape and move to Nairobi:


*“I therefore decided to move out of my grandmother’s and I moved in with this man but he was not a peaceful man he was violent …… He loved fighting and verbal abuse lots of insults…. He used to hold the knife on me and point it towards me and I would just watch.”*
(MF 033)

In Nairobi she became homeless with her child, and started sex work in order to provide for her child and then began to use alcohol to cope with these stressors, which is a similar story to those told by other participants. It was not until she was ill and diagnosed with liver cirrhosis that she stopped using alcohol.

### 3.3. Age and Socio-Demographic Differences in Alcohol and Substance Use

Harmful alcohol and cannabis were more common among younger women < 25 compared to older women > 35 (alcohol: aOR = 0.67; 95%CI: 0.46–0.99; *p* = 0.05; cannabis: aOR = 0.27; 95%CI: 0.17–0.43; *p*-value < 0.001; Table 2; model 1). Moderate/high risk amphetamine use was more common among women in lower SES groups (aOR for highest SES compared to lowest SES: 0.63; 95%CI: 0.42–0.94; *p*-value: 0.009), as well as being more commonly reported among Muslim women (aOR = 2.39; 95%CI: 1.17–4.87; *p* = 0.0008) and married women (aOR = 1.68; 95%CI: 1.05–2.67; *p* = 0.03) in model 1.

Qualitative interviews indicate reasons for age differences in women’s use of alcohol, with older women reporting greater responsibilities for supporting dependents including raising school fees for their children and meeting parental obligations. One example is a 43-year-old mother of three children, who had a first child following early sexual debut at age 15, remarried five years later and gave birth to three more children. She left the man when he repeatedly misused alcohol and physically abused her. She talked about her own alcohol consumption:


*“Mine it does not affect me. You know I drink controllably and I will not drink to the extent that I am so drunk, and it is not every day.”*
(MF 282)

In contrast, heavy alcohol use was observed in the younger sex workers. A 25-year-old woman explained how she is never able to refuse the drinks that clients buy for her:


*I: So…how much do you drink in a given day? R: If it is beer because me I love beer so much; I will take… ten. I: You take ten bottles? R: Yes, that is for the whole night… You are bought by a client there because if you dance well they tell you sit down and be given [a drink], will you refuse?*
(MF 240)

Unlike alcohol and cannabis, women reported that amphetamine is cheap and readily available in many informal settings, and has different effects to alcohol.


*“I: So khat helps. R: It helps. I: And what side effects does it have? R: And again when you chew khat you can’t get drunk.”*
(MF282)


*“Muguka (khat) is very cheap.”*
(MF0569)

### 3.4. Alcohol and Other Drug Use during Sex Work

Women with harmful alcohol use were more likely to report forced sexual debut (aOR = 2.25; 95%CI: 1.62–3.13; *p* < 0.001; Table 3, model 2).

Women with harmful cannabis use were more likely to report selling sex in public places (aOR= 3.26; 95%CI: 1.23–8.65; *p* = 0.03) and more likely to have recently migrated for sex work (aOR= 1.80; 95%CI: 1.20–2.72; *p* = 0.005) in model 2. Women with harmful alcohol use were less likely to report condom use at last sex with clients (aOR= 0.61; 95%CI: 0.42–0.87; *p* = 0.006, model 2). However, women who reported harmful alcohol use had significantly lower HIV prevalence (16.8% vs. 28.0%; aOR = 0.41; 95%CI: 0.27–0.62; *p* < 0.001). There was no association with any STIs. Harmful alcohol and cannabis use were associated with increased odds of recent sexual and/or physical violence from non-intimate partners (alcohol: aOR = 1.60; 95%CI: 1.13–2.27; *p* = 0.008; cannabis: aOR = 2.03; 95%CI: 1.31–3.15; *p* = 0.001; Table 4, model 3a), as well as of recent arrest.

In qualitative interviews, participants explained how alcohol consumption and use of other drugs enabled them to navigate the sex work environment by alleviating their shyness, relaxing them and giving them courage to negotiate with clients. The majority consumed alcohol at the request of clients and rarely used their own money to buy alcohol except for when they sat in bars to wait for potential clients. Alcohol was more commonly used in sex work in comparison to other drugs, but smoking cannabis before sex work was reported, as illustrated below:


*“I don’t drink daily, I take when am going to work because of that fear… to remove that fear.”*
(MF024)


*“Sometimes it [bhang] calms me. Because in this job you face so many challenges… So me when I smoke I feel good, relaxed, and if I don’t smoke when going to the streets I will not work.”*
(MF0569)

Women reported risks associated with alcohol use during sex work including difficulty in negotiating condom use, being vulnerable to physical and sexual violence, risky sexual behaviours, and loss of income. One woman recounted how due to her alcohol overuse she had become dependent on Post-Exposure Prophylaxis (PEP) and it took the intervention of the SWOP clinic to help her reduce her alcohol use:


*“There is a time I used to drink that is when I was at risk. Because you would find that you go to work drunk and to negotiate with a client about the condom will be a bit tough because yourself you are drunk, you find that you just had sex without the condom. So those days I was on PEP a lot but I was advised that it is either I reduce alcohol or I stop [drinking] completely.”*
(MF 012)

Women reported how drunken clients threatened violence when they tried to negotiate condom use, leading to women relying on PEP or Pre-Exposure Prophylaxis (PrEP) for HIV prevention. The following excerpt describes women’s coping with intoxicated violent clients:


*“I… another one his drunkenness is showing him [in] fights. So you see instead of fighting just do what brought you there [sex work], and the next day you know what to do. Me that is when I go [to] SWOP to pick PEPs….”*
(MF 255)

Some women reported stopping or moderating their alcohol use to improve their ARV adherence, including after advice from SWOP clinics:


*“I used to get so drunk… I had to tell even the people from SWOP and they told me to reduce. Because…the drugs [ARVs] will not work you will even forget to take.”*
(MF497)

Amphetamine use was not associated with risky sexual behaviours or violence according to quantitative or qualitative results (Table 4, model 3a).

### 3.5. Family Relationships, Mental and Physical Health

According to quantitative analysis, harmful alcohol use was associated with increased odds of mental health problems including PTSD (aOR = 3.20; 95%CI: 2.00–5.12; *p* < 0.001) and depression/anxiety (aOR = 2.36; 95%CI: 1.58–3.52; *p* < 0.001) in model 3b (Table 4). Women with harmful cannabis or amphetamine use were also more likely to report increased odds of depression/anxiety in adjusted analysis.

In the qualitative interviews, women reported how alcohol and cannabis use had negative effects on their relationships particularly with children. One woman who has now stopped consuming alcohol and other drugs recounted how drinking had affected her relationship with her children:


*R: “I used to get so drunk that even when my children see me they would start crying.... I had to tell even the people from SWOP and they told me to reduce. Because these children of yours first you will give them stress, second they will join that behaviour you are showing them… Now that is where I started because I was like at times when I get to the house and you ask me a question I will respond with a kick.”*
(MF 497)

Physical health complications following cannabis use were also reported by FSWs including addiction/dependence:


*I: “Is there a possibility that one can become an addict? R: I literally smoke bhang everyday whether I am going to work or not even while in the house I have to smoke bhang.”*
(MF132)

In the qualitative interviews, six women reported becoming abstinent from alcohol and other drugs for health reasons, often after medical advice from the SWOP clinic staff.

## 4. Discussion

We found a high prevalence of harmful alcohol and other drug use, as well as poly-substance use, among FSWs. Childhood neglect and violence were drivers of entry into sex work, and there were strong associations between ACEs, in particular street homelessness in childhood, and harmful substance use. Alcohol and cannabis were commonly used during sex work, and both were associated with recent violence from clients; however only alcohol was associated with risky sexual behaviours such as reduced condom use. Women reported negative effects of harmful alcohol and cannabis use including dependence, mental and physical health concerns, and impacts on their relationships with their children. In contrast, amphetamine use was more common among married Muslim women in lower SES groups, was not used during sex work and was not associated with violence or risky sexual behaviours. All substances increased the risk of mental health problems.

The prevalence of reported harmful alcohol use in our study was slightly higher than previous studies among FSWs in Kenya (~20% using the AUDIT tool [16,23,24]). It was higher than estimates for women in the general population (prevalence of heavy episodic drinking among women in Kenya: 2.5%, daily alcohol consumption: 12.7% [49]), although national estimates do not employ a validated alcohol-use tool so comparisons are limited. Data from the 2012 Kenyan national drug and substance use rapid assessment indicates that the prevalence of non-alcohol drug use was much higher among FSWs in our study compared with in the general population [50]. The majority of amphetamine use reported by women in our study was ‘mira’ or ‘khat’ use, an amphetamine-like plant which is legal, locally grown and one of the most popular drugs in Kenya [51]. The health effects of mira are less well-studied than those of cannabis and alcohol, but some studies have shown possible dependence with heavy use [52] and increased risk of psychosis [51,53]. Future interventions should consider addressing poly-substance use, given how frequently women reported harmful use of multiple substances. Of note, there was almost no reported recent injecting drug use among women in our study. Lifetime reported injection drug use was also low, which is in contrast to previous research in Kenya indicating an increase in heroin use in the general population [28] and among sex workers; for example, a prevalence of 49% lifetime injecting drug use was reported among FSWs in Kisumu [18]. This may be due to under-reporting of injection drug use in our study, or may reflect geographical differences in drug use within Kenya.

It is important to consider FSWs’ alcohol and other drug use through a syndemics framework, as the co-occurrence of health and social risk factors which perpetuate each other across the life course suggests the need to address the interconnected nature of these risks. FSWs in our study had a high prevalence of ACEs, including experience of childhood physical and sexual violence, and street homelessness, which were strongly associated with current harmful alcohol and other drug use. These experiences were reflected in qualitative interviews with women reporting childhood neglect, violence, homelessness and poverty, which were often drivers for entry into sex work and alcohol use. Childhood experience of violence and trauma—including multiple ACEs [21]—has been shown to increase the risk of harmful alcohol and other drug use [54] and the risk of revictimisation and violence in adulthood [55] among women in the general population. Our findings suggest that interventions to address harmful alcohol and other drug use should tackle upstream structural drivers including homelessness, ACEs, violence and poverty. Previous research from Kenya has found that community health workers can provide support to children who experience sexual abuse, by helping them to report the abuse to police, and to access healthcare services and alternative housing [56]; therefore, improved training of community and other healthcare workers is a key area for policy makers to address. An increasing number of interventions have been developed in Sub-Saharan Africa addressing parent–child relationships and child maltreatment, many focusing on tackling socio-cultural gender and sexual health norms. For example, in Rwanda, the MenCare+ programme involved men in child health, couple communication, and violence prevention initiatives, and the ‘Parenting for Respectability’ intervention with fathers and mothers aimed to address alcohol and violence in Kampala, Uganda [54,55]. Alcohol and cannabis use were also associated with food insecurity; community and family programmes need to be accompanied by structural interventions addressing poverty, including improved access to stable employment, financial support, housing and education.

Alcohol use is embedded in the sex work environment, with women reporting alcohol use to cope with sex work and due to pressure from clients. Its association with risky sexual behaviours including reduced condom use [15,16,23] has been commonly reported among FSWs in Sub-Saharan Africa [8,16,57]. We found that cannabis was also used by women during sex work, but unlike alcohol its use was not facilitated by clients and was not associated with reduced condom use. Alcohol and cannabis use were associated with violence from clients and police arrest, which was also reflected in the qualitative interviews. Interventions addressing violence have been shown to be effective among women in LMICs [58,59] as well as among FSWs [60], and these should be integrated into future alcohol use interventions, for example by screening women with alcohol use problems for experiences of violence, and vice versa, and referring them for appropriate support. Although their use was also prevalent, amphetamines were not widely used during sex work and did not increase risky sexual behaviours. Women’s risk perception of alcohol and other drug use was that it increased their risky sexual behaviours and the risk of violence during sex work; many reported trying to moderate their alcohol use in order to protect themselves, and accessed PEP and PrEP when needed. Alcohol, cannabis and amphetamine use were all associated with increased risk of depression/anxiety among women in our study, which echoes findings from a recent systematic review on the mental health of FSWs in LMICs [20]. This association is considered to be bi-directional—harmful alcohol and other drug use has been shown to increase the risk of mental health problems [61], while mental health problems can drive women to use substances as coping mechanisms [62]. There are currently no published mental health interventions for FSWs—future research should consider integrating harmful alcohol and drug use interventions within mental health services.

Interventions among FSWs in Kenya [63] and Mongolia [64] indicate that brief alcohol reduction interventions can be effective in reducing alcohol use and binge drinking among FSWs. In addition, a drug use intervention programme among FSWs in Durban, South Africa, reported reductions in drug use and numbers of sexual partners [65]. The few published alcohol and drug use interventions for FSWs in LMICs have mainly focused on addressing alcohol and drug use at the individual level, and only one used a validated tool to report on alcohol use, which affects the validity of the reported results. The ASSIST tool is an up-to-date validated tool for assessing alcohol and drug use, which is recommended by the WHO [36], links directly to treatment approaches, and was easy to administer by staff in our study. Future studies with FSWs should consider the importance of measurement tools, as to our knowledge this is the first study among FSWs in LMICs to use a validated tool to measure substance-specific drug use other than alcohol. A major barrier to providing care for individuals with alcohol and substance use problems is the lack of availability of contextually appropriate treatment programmes, with a large treatment gap particularly in LMICs. Evidence-based brief interventions such as the Counselling for Alcohol Problems (CAP) programme have been shown to be beneficial for treating alcohol use problems among primary care populations in India [66]. A simple intervention such as the CAP programme can be delivered by non-specialists and has the potential to be upscaled in low-resource settings and adapted to local contexts such as for FSW populations. Given the availability of alcohol and other drugs and pressures on FSWs to use these during sex work, policy makers designing interventions should consider broader structural, environmental and economic factors. Social and cultural norms have a major influence on alcohol and drinking patterns. Research from LMICs indicates that policies which affect the availability of alcohol and legal drinking age are predictors of alcohol consumption [67]. Advocating for policy changes around alcohol pricing, accessibility and advertising is crucial, as these factors strongly impact alcohol consumption [4,68,69,70]. Interventions should focus on addressing social norms around alcohol use, and improving the overall safety and autonomy of women engaged in sex work. Finally, researchers and policy-makers should consider how the criminalisation of sex work [7] makes it difficult to regulate and improve the safety of the sex work environment. Globally, increased criminalisation of sex work is linked to worse health and social outcomes [7], and advocating for laws which improve the safety of sex workers is key to improving alcohol- and drug-related risks.

Our findings that harmful alcohol use was associated with a reduced HIV prevalence are in contrast to previous studies on alcohol use and HIV among FSWs [16,71]. However, this finding was explained by our qualitative data, with women reporting reducing their alcohol consumption on advice from SWOP clinics. This suggests that HIV services can address risky behaviours, and adds to growing evidence [71] of the importance of embedding alcohol and other drug use interventions within HIV/STI services particularly for key populations.

### Strengths and Limitations

A major strength of our study was the use of validated tools to assess alcohol and other drug use, ACEs and violence, and qualitative interviews which were used to inform our conceptual pathways and triangulate our findings. Approximately 50% of the estimated FSW population in Nairobi are in active follow-up at one of the seven SWOP clinics, from where we drew our sample. Due to the stigmatised nature of sex work it is possible that the most vulnerable women may have been under-represented (as they had not visited a SWOP clinic). Future research in Nairobi should consider actively recruiting women engaged in sex work who are not members of SWOP clinics, to ensure all women are represented. Ongoing efforts by peer educators to identify and recruit new FSWs to SWOP clinics and to reduce stigma will also be crucial in ensuring the most vulnerable women are engaged. There is potential for recall bias and under-reporting of sensitive topics including ACEs, alcohol or other drug use, and violence. For exposures such as ACEs, we can generally assume the direction of association, however it is less clear for more proximal risk factors including risky sexual behaviours and recent violence. Longitudinal data will become available as the study progresses, enabling better understanding of the direction of association between exposures such as violence and alcohol and other drug use.

## 5. Conclusions

Our study found a high prevalence of harmful alcohol, cannabis, amphetamine and polysubstance use among FSWs. Quantitative and qualitative results indicate differing and overlapping risks for substances. Alcohol and cannabis use are associated with more risks in the sex work environment, including alcohol use with reduced condom use and alcohol and cannabis use with recent violence from clients, while amphetamines were not associated with risks during sex work. All substances are associated with street homelessness in childhood and recent poor mental health, while harmful alcohol and cannabis use were associated with a higher Adverse Childhood Experience (ACE) score overall. Our findings highlight key syndemic risk factors across the life course, starting in early life. They indicate the need for individual and structural level interventions tailored for FSWs addressing harmful alcohol and drug use.

## Figures and Tables

**Figure 1 ijerph-19-07294-f001:**
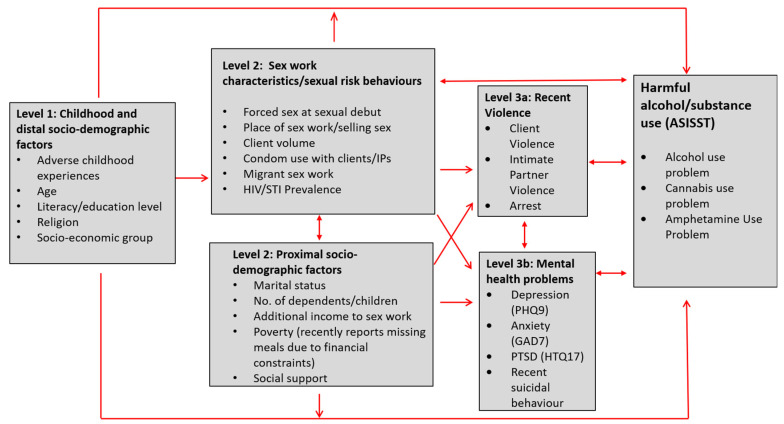
Conceptual framework for exploring risk factors for harmful alcohol and substance use.

**Figure 2 ijerph-19-07294-f002:**
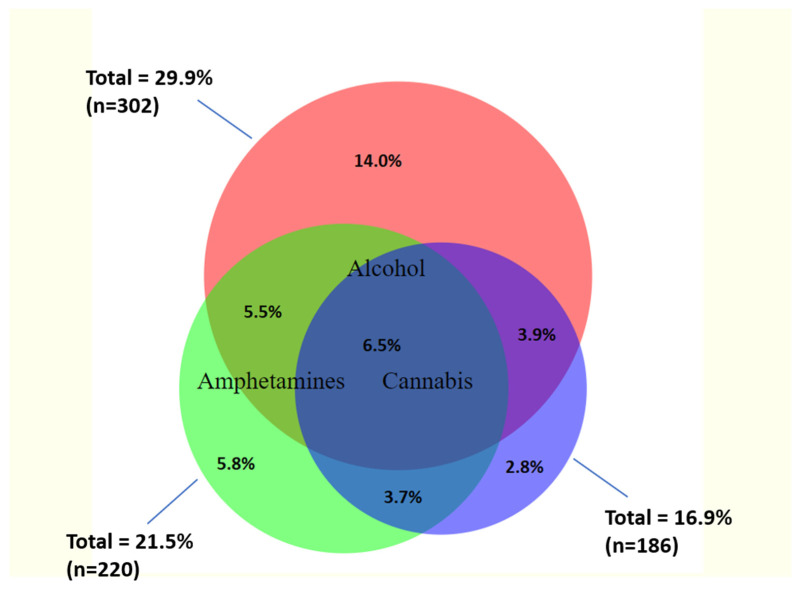
Prevalence and overlap between harmful alcohol, amphetamine and cannabis use among FSWs in Nairobi.

**Table 1 ijerph-19-07294-t001:** Study sample characteristics and associations with harmful alcohol and other substance use.

			N (%) (N = 1003)	Harmful Alcohol Use % (N = 304)	Harmful Cannabis Use % (N = 186)	Harmful Amphetamine Use % (N = 220)
*Level 1: Adverse Childhood Experiences and distal socio-demographic characteristics*						
Age (years)	<25		212 (11.7%)	34.1	33.5	26.9
	25–34		353 (39.4%)	35.9	21.0	25.3
	35+		438 (48.9%)	23.8	9.5	17.1
Adverse childhood experiences	Ever lived on the streets	no	878 (88.0%)	27.5	14.2	19.3
		yes	125 (12.0%)	46.5	36.1	37.5
	Experienced physical/sexual violence	no	212 (20.7%)	14.0	8.6	15.9
		yes	791 (79.3%)	33.9	19.0	22.9
	Total number of ACEs reported	<4	281 (28.7%)	17.7	9.1	16.9
		5–8	540 (54.1%)	31.5	17.9	21.3
		9–12	170 (17.2%)	44.9	25.6	28.4
Literacy	Illiterate		166 (17.6%)	25.1	13.6	17.2
	Literate		837 (82.4%)	30.8	17.5	22.4
Religion	Catholic		375 (36.9%)	35.0	17.7	23.8
	Protestant		534 (54.4%)	25.7	14.2	17.2
	Muslim		46 (4.6%)	34.2	20.8	41.5
	Other/none		46 (4.6%)	31.0	37.8	35.1
Socio-economic Status (SES)	Low/low-middle		401 (39.3%)	33.8	18.2	25.6
	middle		200 (19.9%)	23.9	19.6	26.1
	Upper middle/upper		400 (40.8%)	29.0	14.2	15.2
*Level 2: Proximal socio-demographic characteristics, sex work characteristics, sexual risk behaviours, HIV and STI prevalence*						
Marital status	Ever married	No	216 (18.8%)	28.4	19.6	17.0
		Yes	787 (81.2%)	30.1	16.2	22.5
No. of children ^1^	None		40 (3.4%)	41.3	32.7	18.9
	1–2		644 (66.1%)	30.1	16.6	20.9
	3+		264 (30.5%)	28.2	16.3	24.2
Income	No additional income		571 (56.4%)	28.9	18.1	24.0
	Income other than sex work		432 (43.7%)	30.9	15.2	18.1
Hunger	Skipped a meal in the last 7 days due to financial constraints	No	670 (66.1%)	25.4	14.2	18.8
		yes	331 (33.9%)	38.5	22.0	26.7
Social support	Someone to talk to about your problems	No	278 (27.5%)	31.8	18.2	23.3
		yes	725 (72.5%)	29.0	16.3	20.8
Age first sex	Mean (years)		16.3	16.1	15.6	15.9
Age first sex work	Mean (years)		24.4	23.3	21.5	22.8
Forced sexual debut	Tricked, pressured or forced into sex	No	695 (68.7%)	24.2	15.6	21.6
		yes	306 (31.3%)	42.1	19.6	21.2
Place of selling sex	Phone/internet/friends		54 (5.4%)	25.5	8.2	19.0
	Home/middle men/markets		15 (1.6%)	21.5	3.5	32.1
	Brothel/escort service/massage parlour		14 (1.5%)	7.7	7.7	30.8
	Bar/club/lodge/social gatherings		620 (61.5%)	30.3	15.9	18.8
	Street/bus/taxis		294 (30.0%)	30.3	21.5	26.8
Place of sex work	Lodge/hotel/rented room		907 (91.4%)	29.0	16.5	21.6
	Other public place		28 (2.8%)	50.0	41.9	30.0
	Home		60 (5.8%)	30.1	10.5	17.3
Client volume/week	Median		3	4	4	4
Condom use last sex with intimate partner	Last sex with IP	No	411 (65.8%)	31.5	16.2	22.4
		yes	199 (34.2%)	25.6	13.4	15.7
Condom use at last sex with client		No	200 (19.8%)	41.0	20.3	26.1
		Yes	784 (80.2%)	27.2	16.3	20.8
Migrant sex work	Sold sex outside Nairobi in last 6 months	No	730 (73.0%)	28.0	14.3	21.5
		yes	264 (27.0%)	34.2	24.0	22.0
HIV status		Negative	746 (72.0%)	34.4	19.4	24.1
		Positive	257 (28.0%)	17.9	10.2	14.7
Chlamydia Trachomatis		Negative	932 (94.3%)	29.2	15.7	21.2
		Positive	67 (5.7%)	38.0	34.3	27.0
Neisseria Gonorrhoea		Negative	975 (97.4%)	29.9	16.7	21.5
		Positive	67 (2.6%)	23.4	21.3	21.3
Syphilis (Treponema pallidum)		Negative	979 (97.9%)	30.0	16.9	21.6
		Positive	20 (2.1%)	26.3	15.8	10.5
Bacterial vaginosis		Negative	428 (42.1%)	32.8	17.5	21.5
		Positive	199 (20.0%)	30.2	15.9	19.9
		Intermediate	371 (37.9%)	26.4	16.5	22.2
Trichomonas vaginalis		Negative	969 (97.0%)	29.7	16.6	21.6
		Positive	31 (3.0%)	31.5	24.0	14.8
*Level 3a: Recent violence/arrest (last 6 months)*						
Any recent sexual and/or physical non-IP violence		no	456 (45.0%)	20.3	9.8	16.0
		yes	547 (55.0%)	37.6	22.7	26.0
Any recent sexual and/or physical IP violence		no	693 (69.1%)	24.7	15.5	19.9
		yes	310 (30.9%)	41.1	19.8	24.9
Recent arrest		No	701 (69.3%)	24.4	13.0	16.6
		yes	302 (30.7%)	42.0	25.5	32.7
*Level 3b: Mental health problems*						
Depression	PHQ9 score ≥ 15	No	778 (76.8%)	23.0	14.0	18.1
		yes	222 (23.2%)	52.5	26.4	32.9
Anxiety	GAD-7 score ≥ 10	No	894 (89.9%)	26.3	15.8	20.2
		yes	109 (11.0%)	57.6	25.2	31.3
PTSD	HTQ17 score ≥ 2.5	no	856 (85.8%)	24.8	15.5	20.2
		yes	138 (14.2%)	60.2	25.4	28.6
Recent suicidal behaviour	suicidal ideation or attempt last 30 days	No	902 (89.8%)	27.3	15.4	20.2
		yes	101 (10.2%)	51.6	29.8	32.6

^1^ missing *n* = 55.

**Table 2 ijerph-19-07294-t002:** Multivariable logistic regression—associations with alcohol/substance use and level 1 variables.

Model 1			Crude OR (95%CI)	Adjusted OR * (95%CI)	*p*-Value
Alcohol	Number of reported ACEs	≤4	1.0	1.0	
		5–8	2.14 (1.50–3.04)	2.06 (1.43–2.97)	
		9–12	3.78 (2.47–5.79)	3.66 (2.34–5.72)	<0.001
	Experienced sexual or physical violence in childhood		3.16 (2.09–4.77)	3.27 (2.13–5.02)	<0.001
	Street homelessness as a child		2.29 (1.57–3.36)	1.95 (1.31–2.92)	0.001
	Age	<25	1.0	1.0	
		25–34	1.08 (0.76–1.54)	1.18 (0.82–1.71)	
		35+	0.60 (0.42–0.86)	0.67 (0.46–0.99)	0.05
	Literacy	Literate	1.33 (0.93–1.33)	1.48 (1.00–2.20)	0.05
Cannabis	Number of ACEs	≤4	1.0	1.0	
		5–8	2.20 (1.40–3.45)	1.99 (1.23–3.21)	
		9–12	3.45 (1.05–5.80)	2.99 (1.72–5.21)	<0.001
	Experienced sexual or physical violence in childhood	2.50 (1.54–4.05)	2.58 (1.52–4.37)	<0.001	
	Street homelessness as a child	3.41 (2.27–5.12)	2.61 (1.66–4.10)	<0.001	
	Age	<25	1.0	1.0	
		25–34	0.53 (0.36–0.77)	0.61 (0.41–0.92)	
		35+	0.21 (0.14–0.32)	0.27 (0.17–0.43)	<0.001
Amphetamines	Ever lived on the streets in childhood		2.52 (1.69–3.74)	1.92 (1.27–2.91)	0.002
	SES group	Low/lower middle	1.0	1.0	
		Middle	1.02 (0.70–1.50)	1.15 (0.76–1.74)	
		Upper middle/upper	0.52 (0.37–0.74)	0.65 (0.43–0.98)	0.04
	Literacy	literate	1.39 (0.91–2.14)	1.74 (1.14–2.65)	0.01
	Religion	Catholic	1.0	1.0	
		Protestant	0.66 (0.48–0.91)	0.73 (0.52–1.02)	
		Muslim	2.27 (1.21–4.26)	2.39 (1.17–4.87)	
		Other/none	1.73 (0.90–3.32)	1.87 (0.94–3.73)	0.0008

* Model 1 adjusted for all level 1 variables including age, literacy, religion, SES, ACE score and SWOP clinic. ACEs—Adverse Childhood Experiences, SES—Socio-Economic Status, SWOP—Sex Worker Outreach Programme.

**Table 3 ijerph-19-07294-t003:** Multivariable logistic regression: associations with alcohol/substance use and level 2 variables.

Model 2			Crude OR (95%CI)	Adjusted OR * (95%CI)	*p*-Value
Alcohol	Hunger (skipped a meal in the last week due to low income)		1.84 (1.40–2.43)	1.82 (1.29–2.57)	0.001
	Forced sexual debut		2.27 (1.72–3.01)	2.25 (1.62–3.13)	<0.001
	Condom use last sex with client		0.54 (0.39–0.74)	0.61 (0.42–0.87)	0.006
	HIV positive		0.42 (0.30–0.59)	0.41 (0.27–0.62)	<0.001
Cannabis	Hunger (skipped a meal in the last week due to low income)		1.70 (1.23–2.37)	1.48 (0.98–2.24)	0.07
	Place of sex work	Lodge/hotel/rented room	1.0	1.0	
		Public place	3.65 (1.71–7.77)	3.26 (1.23–8.65)	
		home	0.58 (0.27–1.30)	0.62 (0.29–1.34)	0.03
	Migrated for sex work		1.90 (1.35–2.67)	1.80 (1.20–2.72)	0.005
Amphetamines	Ever married		1.42 (0.96–2.11)	1.68 (1.05–2.67)	0.03
	HIV positive		0.54 (0.37–0.79)	0.52 (0.34–0.79)	0.003

* Model 2 adjusted for all level 1 and level 2 variables (including marital status, no. of children, additional income to sex work, social support, forced sexual debut, condom use at last sex, any STIs, place of selling sex, migrated for sex work). ACEs—Adverse Childhood Experiences, SES—Socio-Economic Status, SWOP—Sex Worker Outreach Programme.

**Table 4 ijerph-19-07294-t004:** Multivariable logistic regression: associations with alcohol/substance use and level 3a and b variables.

**Model 3b ***		**Crude OR ** **(95%CI)**	**Adjusted OR *** **(95%CI)**	** *p* ** **-Value**
Alcohol	Recent sexual/physical violence non-IP	2.37 (1.78–3.15)	1.60 (1.13–2.27)	0.008
	Recent sexual/physical violence IP	2.13 (1.61–2.82)	1.56 (1.12–2.19)	0.009
	Recent arrest (6 months)	2.24 (1.69–2.97)	1.93 (1.37–2.72)	0.001
Cannabis	Recent sexual/physical violence non-IP	2.71 (1.91–3.86)	2.03 (1.31–3.15)	0.001
	Recent arrest (6 months)	2.29 (1.65–3.18)	1.60 (1.08 -2.35)	0.02
Amphetamines	Recent arrest (6 months)	2.44 (1.79–3.32)	2.06 (1.43–2.97)	<0.001
**Model 3b ****		**Crude OR ** **(95%CI)**	**Adjusted OR ** ** **(95%CI)**	** *p* ** **-Value**
Alcohol	Moderate/severe depression and/or anxiety	3.79 (2.80–5.12)	2.36 (1.58–3.52)	<0.001
	PTSD	4.54 (3.14–6.56)	3.20 (2.00–5.12)	<0.001
CannabisAmphetamines	Depression and/or anxiety	2.36 (1.68–3.32)	2.08 (1.32–3.28)	0.002
	Depression and/or anxiety	2.15 (1.55–2.96)	1.95 (1.29–2.96)	0.002

* Model 3a adjusted for level 1, 2 and 3a variables (including recent sexual/physical violence from IP). ** Model 3b adjusted for level 1, 2 and 3b variables (including recent suicidal behaviour and PTSD). ACEs—Adverse Childhood Experiences, SES—Socio-Economic Status, SWOP—Sex Worker Outreach Programme. IP—intimate partner.

## Data Availability

The datasets generated and/or analysed during the current study are not publicly available as the study is still underway. However, the datasets will be available from the corresponding author from June 2023 (2 years after study data collection is completed).

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
