# Peer review of "Harmful Alcohol and Drug Use Is Associated with Syndemic Risk Factors among Female Sex Workers in Nairobi, Kenya"

_ijerph, 2022, doi:10.3390/ijerph19127294_

Round 1
Reviewer 1 Report
As I wrote in comments in the text, I believe that the authors could add some personal considerations to the data collected.

Author Response
Response to Reviewer 1
Please write down the full sentence: adverse childhood experiences
Response: We have amended the abstract to include the full sentence ‘Adverse Childhood Experiences’.
When you write "female sex workers", who do you refer to? assigned females at birth? trans women? biological women? I think this issues should be specificied, even for further assumptions in your article
Response: When we refer to female sex workers, this includes women who were assigned female at birth. Transgender women were not included in the study as their risk profile and experiences are likely to differ from Cisgender women. We have updated our methods to specify this (lines 144-147)
I would start the new paragraph here.
Response: Thank you for this suggestion. We have created a new paragraph in the introduction as suggested.
I think it would be useful for the authors to clarify and argue their hypothesis: what could be the common ground between sex work and drug use? Adverse first life experiences? low socio-economic status? STIs? mental disorders? are these elements connected to each other?
Response: We thank the reviewer for this suggestion. The connection between sex work and alcohol/drug use is hypothesized in the introduction in paragraph 1 (lines 50-55) where we discuss the use of alcohol and drugs during sex work to cope with the challenges of sex work and in paragraph 3 (lines 82-95) in which we link these risk factors together using a syndemics framework. In addition our conceptual framework which forms a key part of our hypothesis (figure 1) links early life experiences, socio-economic status, sexual risk behaviours, sex work and alcohol/drug use.
The figure might be clearer.
Response: We have made the figure clearer by adding in frequencies and total percentages for each substance.
Are these your assumptions based on the interviews or did the participants specify that these are the reasons why they entered into sex work?
Response: In our methods section (line 218-224) we state that ‘qualitative interviews were conducted using semi structured guiding questions including background characteristics and entry into sex work, experiences of alcohol and other drug use, experience of violence, mental health and financial situation.’ Therefore women were specifically asked about reasons for entry into sex work. These are not assumptions and are based on participants reported experiences. We have made a minor amendment in the results (lines 373-376) to make it clear how adverse early life experiences including teenage pregnancy, relationship breakdowns, poverty and a need to provide for children drove women into sex work. Although the focus of our manuscript is not on reasons for entry into sex work, this is reported in more detail in a qualitative baseline paper from the Maisha Fiti Study and similar findings are reported in this paper in terms of reasons for entry into sex work.(1).
You are reporting your results, but I think it would be interesting to know your personal assumptions. What are in your opinion, also based on the data collected, the reasons why these people use substances? can you argue your findings? You also investigated recent arrests, physical/sexual violence, anxiety, depression, PTSD in your sample. Can you speculate on the link between these elements and substances use?
Response: We thank the reviewer for this comment and query. We think that our assumptions, which are based on evidence from interpreting our results within the context of available literature, have been explored in depth in the current discussion. The reasons why we think FSWs use alcohol and other drugs are detailed in paragraph 3 (lines 542-568) of the discussion where we detail how ACEs, violence and poverty are drivers of alcohol and drug use and frame this within a syndemics perspective. In paragraph 4 (lines 569-592) we discuss how the sex work environment encourages and normalises alcohol use. We have evidenced these arguments with references to the currently available literature on ACEs, violence, poverty and alcohol use as well as previous research on alcohol use and sex work. We have discussed in paragraph 3 how childhood experience of violence and trauma – including multiple ACEs - have been shown to increase the risk of harmful alcohol and other drug use and the risk of revictimisation and violence in adulthood (lines 549-555). We have discussed a recent systematic review on mental health among FSWs which reflects our findings on alcohol use and mental health and have added an additional sentence in our discussion explaining the possible bi-directional links between mental health and alcohol/drug use (line 584-592).
- Wanjiru R, Nyariki E, Babu H, Lwingi I, Liku J, Jama Z, et al. Beaten but not down! Exploring resilience among female sex workers (FSWs) in Nairobi, Kenya. BMC Public Health. 2022;22(1):965.
Reviewer 2 Report
Thank you for inviting me to review this work. The time spent on its creation and submission is greatly appreciated. The topic is interesting and the study tries to collect different and numerous factors and variables. Although the large number of factors, together with the use of mixed methods, in my opinion is highly ambitious, it being preferable to carry out simpler and more forceful studies, the subject is of great relevance. So here are my recommendations. I hope they are useful:
-The title is too long: it should not exceed 18 words.
-The summary must be balanced. You should dedicate a little more space to the introduction and the objective of the work. In addition, the acronyms in the abstract must appear with the full name the first time (eg: ACEs)
-In the introduction, it is not necessary to indicate the statistical data of the studies consulted (lines 56-60).
-What risk variable is literature? I don't quite understand what they mean by this.
-The first time acronyms appear, the full name must be indicated. For example: SES
-That the objective is to associate variables is closely linked to the quantitative field, but it is far from what can be achieved from the qualitative part of the work. Change the objective of your work or divide it in two, in accordance with the methodologies used.
-Include the reliability of the instruments in your study.
-The qualitative methods used need a description in greater detail. How were the interviews, how long did they last…
-Include a procedure section where you describe in detail the process of your study.
-In the qualitative data analysis, was an analysis program used? When was the saturation point reached?
-In the results section, within the tables indicate the frequency next to each percentage, and vice versa. Similarly, check the percentages in table 1: they do not agree.
-Regarding figure 1, it would be advisable to include the frequency.
-In table 2 and 3, report the acronyms at the bottom of the table. And the significativity
-In the conclusions, place greater emphasis on the practical contributions of the work.
Author Response
Response to Reviewer 2:
Thank you for inviting me to review this work. The time spent on its creation and submission is greatly appreciated. The topic is interesting and the study tries to collect different and numerous factors and variables. Although the large number of factors, together with the use of mixed methods, in my opinion is highly ambitious, it being preferable to carry out simpler and more forceful studies, the subject is of great relevance. So here are my recommendations. I hope they are useful:
Response: We thank the reviewer for their helpful feedback and comments. We hope that the following responses help to clarify and strengthen the manuscript. We understand that the use of mixed methods creates a lengthier and more complex paper, however we feel strongly that the use of both quantitative and qualitative data is essential for exploring the nuances of alcohol and drug use among Female Sex Workers (FSWs) in a way that a single method alone would not capture fully.
-The title is too long: it should not exceed 18 words.
Response: We have shortened the title to address this, and hope that the shorter title is improved and more succinct (lines 2-4). The new title is: ‘Harmful Alcohol and Drug use is associated with syn-demic risk factors among Female Sex Workers in Nairobi, Kenya’. In order to ensure the methods of the study are reflected in the title page and for potential readers we have included ‘quantitative methods’ and ‘qualitative methods’ in the keywords.
-The summary must be balanced. You should dedicate a little more space to the introduction and the objective of the work. In addition, the acronyms in the abstract must appear with the full name the first time (eg: ACEs)
Response: Thank you for your feedback on the abstract. We have edited and added to the introduction of the abstract (lines 18-23) with more detail on the aims. The acronym Adverse Childhood Experiences (ACEs) (line 32) has been written in full.
-In the introduction, it is not necessary to indicate the statistical data of the studies consulted (lines 56-60).
Response: We think it is important to give an overview of the currently available evidence on alcohol and drug use among FSWs globally in order to set the context and frame the scale of the problem. However, in order to improve the readability of the introduction and make it more succinct we have provided the percentages quoted in the studies but removed the confidence intervals (lines 63-65).
-What risk variable is literature? I don't quite understand what they mean by this.
Response: Literacy refers to whether women are able to read and is therefore a marker of education level. It was assessed by asking women during the behavioural questionnaire to read a sentence. The interviewer then recorded whether they were able to read the sentence (yes, partially or no). We have provided a more detailed explanation of this variable in the methods (lines 261-263).
-The first time acronyms appear, the full name must be indicated. For example: SES
Response: Thank you for notifying us of this. We have checked acronyms and ensured they are written in full when first used (line 261).
-That the objective is to associate variables is closely linked to the quantitative field, but it is far from what can be achieved from the qualitative part of the work. Change the objective of your work or divide it in two, in accordance with the methodologies used.
Response: We thank the reviewer for their feedback on our objectives. We have edited our aims and objectives in the introduction (lines 110-114) and methods (lines 229-232) to better reflect our methodologies by providing separate objectives for quantitative and qualitative work. Although our aim overall is to take an integrated approach we understand that the objectives need to separate out what can be achieved by the quantitative work vs. the qualitative work so we hope this change addresses the issue.
-Include the reliability of the instruments in your study.
Response: We have edited the methods to include the reliability of the tools for ASSIST, ACE-IQ, PHQ-9, GAD-7 and HTQ17 (line 170-202).
-The qualitative methods used need a description in greater detail. How were the interviews, how long did they last…
Response: Thank you for your feedback on this – we have edited the methods to include a more detailed description of the qualitative methods (lines 278-299) and the length of the interviews (line 222).
-Include a procedure section where you describe in detail the process of your study.
Response: Our understanding of study procedure is that this refers to a description of how data was collected. The study procedure is detailed in our current methods which have been updated and include extensive details on data collection (line 168-224), as well as two Maisha Fit study baseline papers (one quantitative and one qualitative) which have both now been referenced (line 165) for further details.
-In the qualitative data analysis, was an analysis program used? When was the saturation point reached?
Response: Please see edits and additions to our paragraph on qualitative analysis (lines 278-299) which addresses the issue of saturation (please note in particular lines 286-288 re: identification of emerging themes which is used in preference to the term saturation) and details the analysis program used (NVivo 12) (line 292).
-In the results section, within the tables indicate the frequency next to each percentage, and vice versa. Similarly, check the percentages in table 1: they do not agree.
Response: We thank the reviewer for this suggestion. However, we have given frequencies in column 1, table 1 and in the column headings. Given the volume of data in the table and that the total frequencies are available in the column headings we think that the table will become too busy and difficult to read if these are included as well – this was the process we followed for tables in our baseline mental health paper.
In column 1 of table 1, the reason that the percentages may not match the frequency is that we have used exact frequencies but weighted percentages due to the dataset being weighted and analysed using svyset commands in STATA to account for women <25 being over-represented (please see methods for further details). In table 1 columns 2-4 show row percentages, i.e.it indicates the number of women with the outcome (i.e. harmful alcohol use) in each exposure category e.g. for table 1 among women <25 the proportion of women who have an alcohol use problem is 34.5% while among older women >35 it is 23.8%. This enables us to show how harmful alcohol use (the outcome) varies by exposure category.
-Regarding figure 1, it would be advisable to include the frequency.
Response: We have included the total frequency and percentage for each overall substance category. We hope that this makes the figure clearer for readers.
-In table 2 and 3, report the acronyms at the bottom of the table. And the significativity
Response: We have included the full text of the acronyms at the bottom of tables 2-4. In terms of significance, the P-values are already reported in column 6 of tables 2-4.
-In the conclusions, place greater emphasis on the practical contributions of the work.
Response: We thank the reviewer for this suggestion. We would like to firstly signpost to aspects of the discussion which discuss the practical implications of the work including:
- Reporting on the strong associations with ACEs, which has not previously been reported in any studies with FSWs, and citing and suggesting a number of approaches to address this including childhood and family interventions (paragraph 3, discussion)
- The importance of addressing FSWs health and social outcomes from a syndemics approach, which to date has been limited, rather than addressing these issues in silos.
-the need for validated tools to be used to measure alcohol and drug use, such as the tool used in our study
In addition to this, in response to the above, we have significantly expanded on the discussion with additional suggestions for how alcohol and drug use interventions can be integrated into violence (line 576-579) and mental health (line 584-592) interventions, the need for interventions addressing poly-substance use (line 534-535) as well how to consider addressing wider structural, environmental and economic factors) such as alcohol availability in the sex work industry, and criminalisation of sex work(line 598-630).
Reviewer 3 Report
The authors have presented an interesting investigation of factors related to problem alcohol and substance abuse among sex workers in Nairobi. In essence, the authors have capitalised on the baseline collection of data for what appears to be a longitudinal study of female sex workers, substance abuse and sexually transmitted diseases. While none of the results presented are necessarily novel, it is an interesting and useful investigation.
My key concern for this paper lies in the "So what" of the investigation. While lines 477 to 502 provide a brief overview of the practical implications of this study, the current presentation is largely a theoretical exercise. This investigation adds to the large number of studies that exist that describe the marginalisation of female sex workers and their subsequent increased risk of a host of negative health and social outcomes. The question then becomes, if we know this to be the case, what can we do about it?
Lines 537 to 546 address the stigmatised nature of female sex work, but only from the perspective of limiting access to possible study members. However, within this section the authors also acknowledge that the most vulnerable have not visited a SWOP clinic. Given the value-add from SWOP workers (see line 530 as an example), how can attendance be enhanced?
The authors have obviously invested a significant amount of time in ensuring a well-designed investigation with the use of validated tools. It appears that this investigation has been developed to ensure robust and valid data. Given this, equal consideration should also go into the application of the findings of this investigation.
Author Response
Response to Reviewer 3
The authors have presented an interesting investigation of factors related to problem alcohol and substance abuse among sex workers in Nairobi. In essence, the authors have capitalised on the baseline collection of data for what appears to be a longitudinal study of female sex workers, substance abuse and sexually transmitted diseases. While none of the results presented are necessarily novel, it is an interesting and useful investigation.
My key concern for this paper lies in the "So what" of the investigation. While lines 477 to 502 provide a brief overview of the practical implications of this study, the current presentation is largely a theoretical exercise. This investigation adds to the large number of studies that exist that describe the marginalisation of female sex workers and their subsequent increased risk of a host of negative health and social outcomes. The question then becomes, if we know this to be the case, what can we do about it?
Response: We thank the reviewer for taking the time to read our paper and for this feedback. We would like to signpost to aspects of the discussion which we feel discuss the practical implications of the work including:
- Reporting on the strong associations with ACEs, which has not previously been reported in any studies with FSWs, and citing and suggesting a number of approaches to address this including childhood and family interventions (paragraph 3, discussion)
- The importance of addressing FSWs health and social outcomes from a syndemics approach, which to date has been limited, rather than addressing these issues in silos.
- The need for validated tools to be used to measure alcohol and drug use, such as the ASSSIST tool used in our study. This was the first study among FSWs in LMICs to used a valid tool to measure substance-specific drug use other than alcohol.
In addition to this, in response to the above, we have significantly expanded on the discussion with additional suggestions for how alcohol and drug use interventions can be integrated into violence (line 576-579) and mental health (line 584-592) interventions, the need for interventions addressing poly-substance use (line 534-535) as well how to consider addressing wider structural, environmental and economic factors) such as alcohol availability in the sex work industry, and criminalisation of sex work(line 598-630).
Lines 537 to 546 address the stigmatised nature of female sex work, but only from the perspective of limiting access to possible study members. However, within this section the authors also acknowledge that the most vulnerable have not visited a SWOP clinic. Given the value-add from SWOP workers (see line 530 as an example), how can attendance be enhanced?
Response: Thank you for this feedback. We would like to respond to this comment by providing more detail on the nature of SWOP clinics. The Sex worker outreach program (SWOP) was established in 2008 in Nairobi with the aim of scaling up accessible HIV prevention and treatment services to key populations including FSWs. To enhance accessibility, SWOP clinics were established across seven sites throughout Nairobi, in close proximity to areas with high concentrations of hot spots (areas in which sex workers gather to find clients). Peer sex workers and sex worker community based organisations are integral to the running of SWOP clinics as they engage with FSWs in hot spots where sex is sold. FSWs are recruited into SWOP using a peer support model. Peer educators and outreach workers are current FSWs who have developed skills in capacity building and participation in SWOP. They are seen as leaders within their communities. Outreach workers and peer educators recruit FSWs through health education and facilitate demand creation amongst their peers. Further details of how SWOP clinics recruit FSWs through peer education and micro-planning has previously been reported in detail (1) and we have now referenced this in our methods (line 126-127). As part of the ongoing work of peer sex workers, efforts are continuously made to engage sex workers, including those most at risk, identify new hotspots and ensure attendance is enhanced amongst all sex workers. It is a well-established concern in all research with sex workers and key populations, and we acknowledge that there is always a possibility that some of the most vulnerable are not reached. However, we think that the current model for how SWOP clinics are run in Nairobi aims to reach as many women as possible, including the most vulnerable women. For example, street-based sex workers, who are often considered to be at higher risk of poor health and social outcomes including violence, HIV/STIs etc., are well represented in our sample with over a quarter (30.0%) of our sample reporting soliciting sex from streets/buses and taxis.
We have additionally included a suggestion in our ‘strengths and limitations paragraph’ that ‘Future research in Nairobi could consider actively recruiting women engaged in sex work who are not members of SWOP clinics to ensure all women are represented in the study sample.’(line 650-654)
The authors have obviously invested a significant amount of time in ensuring a well-designed investigation with the use of validated tools. It appears that this investigation has been developed to ensure robust and valid data. Given this, equal consideration should also go into the application of the findings of this investigation.
Response: We thank the reviewer for their feedback and hope that the above responses and revisions in the manuscript address their comments sufficiently.
- Bhattacharjee P, Musyoki H, Prakash R, Malaba S, Dallabetta G, Wheeler T, et al. Micro-planning at scale with key populations in Kenya: Optimising peer educator ratios for programme outreach and HIV/STI service utilisation. PloS one. 2018;13(11):e0205056-e.
Round 2
Reviewer 2 Report
I thank the authors for the effort to improve the manuscript. And your consistent responses. God work!